# Bayés’ Syndrome—A Comprehensive Short Review

**DOI:** 10.3390/medicina56080410

**Published:** 2020-08-13

**Authors:** Emma Murariu, Attila Frigy

**Affiliations:** 1Department of Internal Medicine, Clinical County Hospital Mureș, Gh. Marinescu str. 1, 540103 Târgu Mureș, Romania; emma_murariu@yahoo.com; 2Department of Internal Medicine IV, G.E. Palade University of Medicine, Pharmacy, Science and Technology of Târgu Mureș, Gh. Marinescu str. 38, 540142 Târgu Mureș, Romania

**Keywords:** electrocardiography, P-wave, interatrial conduction block, atrial fibrillation

## Abstract

Prediction and early detection of atrial fibrillation (AF) remain a permanent challenge in everyday practice. Timely identification of an increased risk for AF episodes (which are frequently asymptomatic) is essential in the primary and secondary prevention of cardioembolic events. One of the noninvasive modalities of AF prediction is represented by the electrocardiographic P-wave analysis. This includes the study and diagnosis of interatrial conduction block (Bachmann’s bundle block). Bayés’ Syndrome (named after its first descriptor) denotes the association between interatrial conduction defect and supraventricular arrhythmias (mainly AF) predisposing to cardioembolic events. Our short review presents an update of the most important data concerning this syndrome: brief history, main ECG features, pathophysiological background and clinical implications.

## 1. Introduction

The estimated prevalence of atrial fibrillation (AF) is 3% in the general, adult population of >20 years old, with greater prevalence in the elderly and in patients that are associated with hypertension, heart failure, coronary artery disease, valvular heart disease, obesity, diabetes mellitus and chronic kidney disease. A growth in the number of patients is to be expected in the years to come. AF is recognized as one of the major causes of stroke, heart failure and cardiovascular morbidity in the world. Diagnosis requires its documentation using an electrocardiogram (ECG), with irregular RR intervals and no discernible, distinct P waves, by convention during an episode lasting at least 30 s [1].

As episodes of AF occur frequently asymptomatic and/or underdiagnosed, comporting a high cardioembolic risk, in the last 10–20 years prediction and early detection of AF have become a growing concern in the scientific community. The main ECG-derived variables, proposed and later confirmed in large population-based studies as risk factors and/or markers for developing AF, are presented in Table 1 [2].

## 2. Definition and Short History

Out of all these ECG-derived variables, interatrial conduction block (IAB) has generated a growing interest over time in the medical community.

In 1979, Bayés de Luna published a paper about atrial conduction abnormalities, which were classified as intra- and inter-atrial blocks, the latter being further divided into partial and advanced type [3,4]. In 1985, Bayés de Luna and his team determined that the prevalence of advanced IAB was 0.1%, increasing up to 2% when patients with heart valve disease or cardiomyopathy were considered [4,5]. In 1988, the same author published a new study on patients undergoing 24-h Holter monitoring, showing that patients with IAB had higher rates of supraventricular tachycardia compared to those without advanced IAB; furthermore, advanced IAB was associated with a higher incidence of supraventricular premature beats [4,6]. Agarwal et al. observed a higher prevalence of IAB in new onset AF cases versus a group with sinus rhythm [4,7]; Ariyarajah et al. demonstrated that in patients with probable diagnosis of cardioembolic stroke, those patients with IAB on the ECG had a greater probability of presenting left atrial enlargement or a thrombus in the left atrium [4,8]. The first international consensus report on IAB was published in 2012 [9].

IAB represents a marker and electro-anatomical substrate for the development of supraventricular (atrial) arrhythmias. The association of IAB and supraventricular arrhythmias, particularly atypical atrial flutter and AF, was named (after its first descriptor) Bayés’ Syndrome [4].

## 3. Anatomo-Electrical Background

Intra- and interatrial electrical pathways are represented by four bundles: (1) the Bachmann’s bundle (BB), which is the anterior internodal pathway, having also a (2) branch which connects the right atrium to the left atrium, (3) the Wenckebach’s bundle, which is the middle internodal tract, and (4) the Thorel’s bundle, which represents the posterior internodal pathway [10].

During normal sinus rhythm, the largest, most common and preferential anatomical route for interatrial conduction is through the BB and, consequently, IAB is the result of a conduction delay or complete block in this pathway [11]. It has to be mentioned, that other, more slowly conducting structures also electrically bind the two atria. These are located at the coronary sinus, in the antero-superior interatrial septum and in the postero-inferior interatrial septum [12]. The electrophysiological background of advanced IAB is thought to be a situation when a sinus impulse can no longer pass via the Bachmann region, instead, it propagates towards the AV node depolarizing the right atrium, then the left atrium is depolarized in a caudocranial direction starting from the inferior left atrium, near the atrioventricular node (most frequently the coronary sinus, and in a small proportion the fossa ovalis). This superior–inferior–superior activation pattern is responsible for the biphasic (+/−) aspect of the P waves in the inferior ECG leads (II, III, aVF). The IAB types and their pathophysiological and ECG features are presented in Table 2 [9].

Figure 1, Figure 2 and Figure 3 illustrate the characteristic ECG patterns of first- and third-degree IAB.

Interestingly, procedures such as pulmonary vein antrum isolation could modify the electrocardiographic IAB pattern, up to its disappearance, because of the loss of a part of left atrial signals [13].

## 4. Pathophysiology and Morpho-Functional Substrate

Atrial morpho-functional remodeling, consisting of fibrosis, atrial enlargement and consecutive alteration of atrial function, is responsible for and related to the development of IAB. Atrial fibrosis is the common end-pathway for various cardiac injuries and is able to produce a delay and/or blockage in cardiac electrical conduction. Although the exact role of atrial fibrosis in the development of IAB is still under investigation, it seems that it represents an important structural substrate [11].

It was observed that IAB shows a degree of reversibility: in several studies a reduction of the P-wave duration over time has been noted, indicating the dynamic character of the pathological processes leading to IAB. This phenomenon could be related to reverse atrial remodeling [11].

Atrial cardiomyopathy has been defined as “any complex of structural, architectural, contractile or electrophysiological changes affecting the atria with the potential to produce clinically-relevant manifestations” [14]. It is well known that the development of atrial cardiomyopathy in the setting of atrial fibrillation depends mainly on AF duration: very short-term AF produces no ultrastructural alterations, AF lasting several weeks causes cardiomyocyte changes, while long-term persistent AF produces combined cardiomyocyte and fibrotic changes. Atrial fibrosis plays an important role in the progression of long-term persistent AF to permanent form. Atrial fibrillation-induced complex atrial remodeling is the substrate of the maintenance, progression and stabilization of AF [14].

Left atrial enlargement (LAE) is often associated with IAB, however its presence is not mandatory for the development of IAB [9]. The presence of IAB in an already enlarged LA determines (because of the electrical delay in the activation of the atria) an increased alteration of the LA systolic indices over time. IAB has been proposed as a marker of the structurally altered and electromechanically dysfunctional LA, and, at the same time, as an electrophysiological risk factor and substrate for AF and conditions related to AF, such as congestive heart failure [15,16]. It is important to mention that interatrial dyssynchrony caused by IAB, beyond generating atrial remodeling (increased atrial pressure, atrial dilation and atrial fibrosis), produces endothelial dysfunction, contributing via both factors to enhanced local thrombogenesis, responsible for cardioembolic events [17].

## 5. Imaging Correlations

Attempts to correlate inter-atrial septum thickness (measured by computed tomography) to IAB, in order to use it as a predictor for AF recurrence in patients undergoing catheter ablation for paroxysmal AF, have proven unsuccessful [18]. Rather than assessing the thickness of atrial walls, determining the degree of atrial fibrosis seems to be more beneficial. The atrial fibrotic process in the LA can be assessed by various methods, the gold standard being cardiac magnetic resonance imaging (CMR). Additionally, new echocardiographic techniques, like myocardial deformation imaging (strain, strain rate) using speckle tracking can deliver indirect information about the fibrotic process [19].

Classical transthoracic echocardiography can assess the LA both anatomically and functionally. The representative of the LA size is not its antero-posterior diameter, but its volume, measured by two-dimensional, or—recently—three-dimensional methods. Determining the reservoir function, reflected by the difference between the maximum (before the mitral valve opens) and the minimum volume (when the mitral valve closes), is the most important in this regard [19].

Tissue Doppler Imaging (TDI) can assess the atrial electromechanical delay (EMD), helping to identify patients at risk of developing IAB and Bayés’ Syndrome. EMD is defined as the time interval between the onset of atrial electrical activity (onset of P wave on ECG) and the atrial contraction (onset of the A’ wave on TDI). Atrial strain and strain rate, obtained using speckle tracking, are reduced in the case of fibrosis, revealing a decreased atrial compliance and an altered reservoir function, even before the LA starts to dilate [19]. In a 2018 study, IAB has been shown to correlate directly with the structural remodeling and with the decrease in absolute values of LA strain rate [20].

CMR provides multiple anatomical and functional data such as parietal thickness, volumes, and, most importantly, the degree of fibrosis. The latter is obtained by the use of gadolinium, a contrast agent which accumulates in the interstitial space. As a result, conditions that modify the interstitial space, such as fibrosis, can be identified by delayed myocardial enhancement sequences: myocardial fibrotic areas are hyperintense, while healthy regions have a hypointense or null aspect. At the level of atria, fibrotic changes can be seen as fine areas of delayed enhancement, and can be quantified using the Utah classification system (from I to IV) [19]. Another CMR-derived technique that could be used to determine fibrosis is the T1 sequence, which calculates the myocardial T1 relaxation time during a 10-s voluntary apnea. This method has the disadvantage of high cost, the need for a high level of experience and the low availability for current clinical practice [19]. 3D reconstruction of 3T late gadolinium enhancement CMR allows visualization of atrial fibrosis in patients with IAB, showing the involvement of the upper part of the septum, where the BB is situated [17].

The amount of fibrosis identified by these techniques is strongly associated with AF recurrence after pulmonary vein isolation. On the other hand, a larger amount of fibrosis induced at the ablation sites equals a better isolation of the AF foci and less chance of recurrence [17].

## 6. Clinical Relevance

First degree IAB is relatively common, is associated with a higher risk of atrial fibrillation, causing and is related to higher global and cardiovascular mortality; third degree IAB, although less frequent, represents a strong marker of LAE and paroxysmal supraventricular tachyarrhythmias including AF [9].

The importance of Bayés’ Syndrome and the value of IAB in predicting the occurrence of AF (new onset or recurrent) has been investigated in different clinical scenarios and in large population studies. The main data are presented in Table 3 [11,21,22,23,24,25,26,27,28,29].

IAB is common in patients with heart failure and these patients have more frequently new-onset AF, ischemic stroke, higher hospitalization and mortality rates. In patients with valvular heart disease requiring surgery, advanced IAB was independently associated with postoperative AF [17].

Severe obstructive sleep apnea is also a predictor of IAB, which in this case could be reversed in patients receiving treatment with continuous positive airway pressure (CPAP) [17,30].

P-wave duration is positively correlated with age [11]. Aging is associated with the appearance of structural changes in the myocardium, predominantly fibrosis. IAB appears frequently in elderly persons, e.g., being present in half of centenarians, as a result of progressive conduction system degeneration. This phenomenon is associated with an increased prevalence of AF/atrial flutter. Both AF and IAB are recognized markers of worse prognosis, with a reduction in life expectancy, higher occurrence of dementia, and worse perceived health status [31]. Among the risk factors for incident AF (age, sex, heart failure, hypertension, diabetes mellitus, coronary artery disease, chronic kidney disease, sleep apnea), age was the single most influential factor—AF is more prevalent in elderly patients [32].

For a long time, the increased risk of ischemic stroke in patients with IAB was thought to be a consequence of AF episodes. Recent research has shown that IAB is an independent risk factor for stroke, even in patients without supraventricular arrhythmias. IAB has been independently associated with an increased risk of non-lacunar ischemic stroke when considering sex, age or race, or the presence of previous episodes of AF [17].

The centenarians with IAB had higher mortality and higher incidence of dementia [33].

Derived from the thromboembolic risk related to IAB, it has been suggested that IAB may also represent a novel risk factor for acute occlusive mesenteric ischemia, regardless of the occurrence of AF [34].

## 7. Prediction Scores of AF and Stroke Involving the Presence of IAB

Given the electrophysiological, pathophysiological and clinical implications of IAB in AF occurrence and its prediction, the development of electrocardiographic scores for predicting AF and stroke has been a very appealing topic for the medical community researching Bayés’ Syndrome. In this regard, different variants have been proposed in recent studies [17].

Fujimoto et al. found that advanced IAB, P wave dispersion, and duration of AF per month were independent predictors of AF recurrence [35]. These findings could suggest a score based on P wave characteristics for predicting the risk of AF recurrence after electrical cardioversion.

Thromboembolic complications seem to be frequent in patients with IAB and high CHA_2_DS_2_-VASc score, independently of the presence of AF [17]. Furthermore, in patients with IAB, but without a history of AF, high CHADS_2_ and CHA_2_DS_2_-VASc scores could be used efficiently to predict ischemic stroke or transient ischemic attacks [36]. Martinez-Sellés et al. showed that a CHA_2_DS_2_-VASc score ≥ 2, supraventricular ectopic activity (>40 atrial premature beats/hour and/or atrial runs on Holter monitoring) and advanced IAB with P ≥ 160 ms are associated with an increased thromboembolic risk in patients without documented atrial arrhythmias. The same authors suggest considering early anticoagulation therapy in these high-risk patients [37]. IAB and particularly advanced IAB is a strong predictor of stroke in elderly patients. Designing a score that combines different risk factors, including the presence of IAB, could be useful in preventing embolic stroke, especially in the elderly patients, where the risk of stroke and implicitly the cognitive impairment is high, even without documented arrhythmias [17].

In the case of embolic stroke of undetermined source (ESUS), both partial and advanced IAB are frequent; the latter has proven to be an independent risk predictor of AF. Advanced IAB, together with the CHA_2_DS_2_-VASc score and left atrial volume index, might be useful in identifying ESUS patients with advanced atrial disease that could potentially benefit from early oral anticoagulation for secondary prevention [38].

## 8. Conclusions

Bayés’ Syndrome has been attracting attention in the research field and clinical practice for the last decades, due to the continuous efforts made towards predicting and preventing ischemic strokes, which are related to atrial arrhythmias—especially to atrial fibrillation. The presence of IAB reflects the installation of those morpho-functional changes in the left atrium, which represent the substrate of AF. On the other hand, IAB facilitates directly, by delayed impulse propagation, the occurrence of atrial arrhythmias. The data from the literature support the strategy that the presence of advanced IAB could be the trigger for initiating prophylactic anticoagulant treatment in patients without known atrial arrhythmias (especially AF), but at increased risk (e.g., conferred by age) of cardioembolic events.

## Figures and Tables

**Figure 1 medicina-56-00410-f001:**
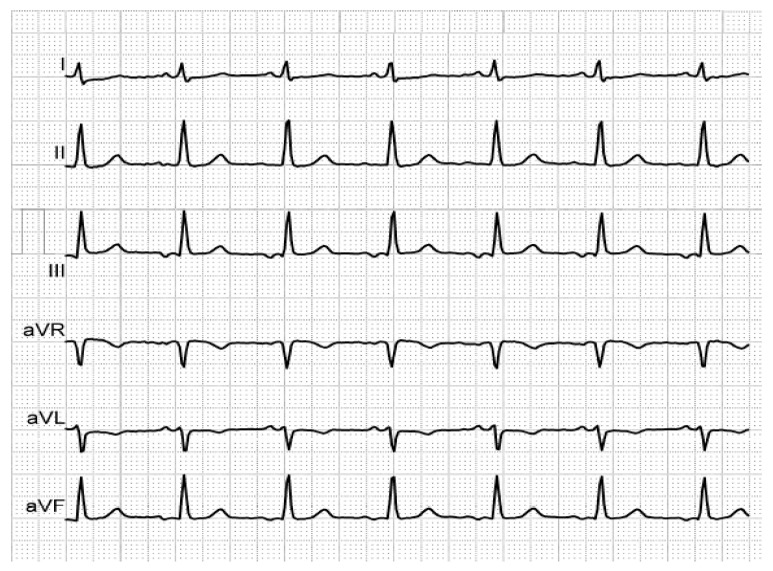
Biphasic P-waves in III and aVF, revealing third-degree IAB, in a patient with mitral prosthesis and episodes of paroxysmal AF.

**Figure 2 medicina-56-00410-f002:**
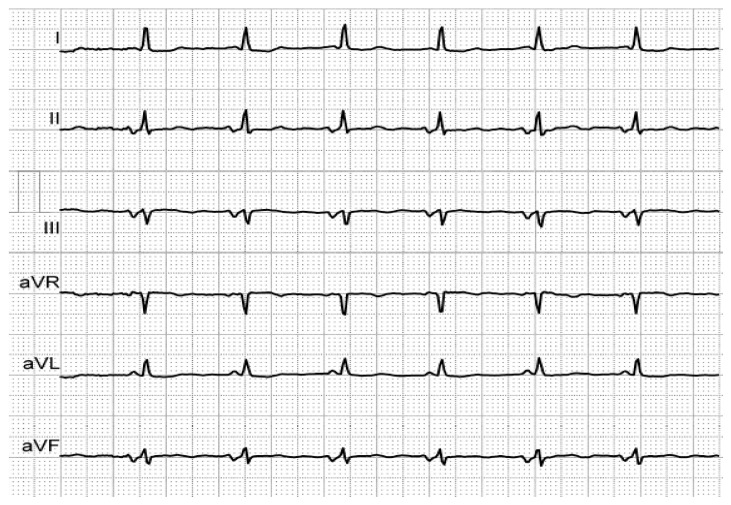
Biphasic P-waves in II, III and aVF, revealing third-degree IAB, in an elderly, hypertensive patient with history of AF.

**Figure 3 medicina-56-00410-f003:**
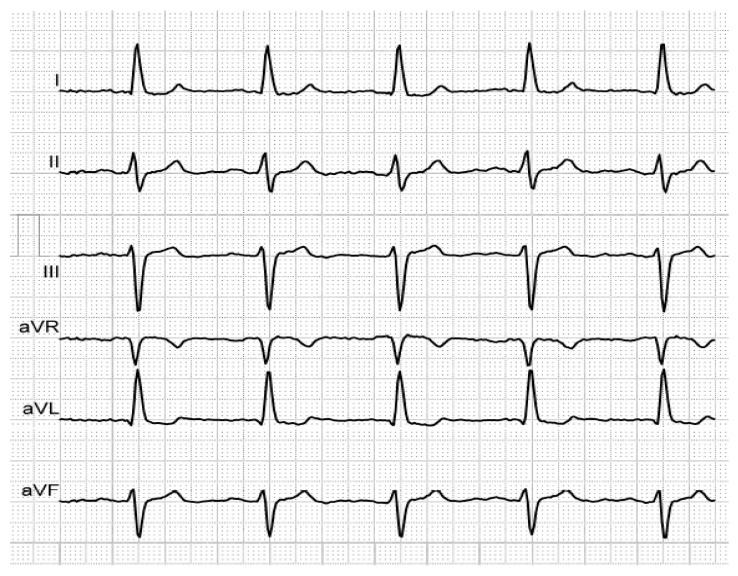
P-wave duration of 140 ms, revealing first-degree IAB, in an elderly, hypertensive patient with history of AF.

**Table 1 medicina-56-00410-t001:** Electrocardiogram (ECG) parameters related to atrial and ventricular electrical activity, with potential role in atrial fibrillation (AF) prediction.

ECG Parameters of Atrial Electrical Activity	Description	Variant for Prediction of AF
average P-wave duration	duration of atrial depolarization	decrease or increase in P-wave duration
maximal P-wave duration	the longest P-wave duration observed on a standard 12-lead ECG	>120 ms
FPD	filtered P-wave duration on the P-wave signal-averaged ECG	≥125 ms
RMS20	the root mean square value of the last 20 ms on the P-wave signal-averaged ECG	≤3.3 μV
PWD	P-wave dispersion = the difference between the longest and shortest P-wave durations on the standard 12-lead ECG	>80 ms
PWSD	the standard deviation of the P-wave durations on the standard 12-lead ECG	>35 ms
IAB	interatrial block = see Table 2	presence
PTFV1	the P-wave terminal force = the product of the negative P-wave deflection in the lead V1 and the duration from the onset of the negative deflection to its nadir	>0.04 mVs
Specific P-wave morphologies	notching or deflection of the P-wave, P-pulmonale	presence
P-wave axis	the direction of the atrial electrical wave-front propagation—altered under a volume or pressure overload of the atrium	an axis outside of 24–74° or <74° in the frontal plane
PACs	premature atrial contractions or runs detected mainly on the Holter ECGs	presence
PR (PQ) interval	Sum of the P-wave and the PR (PQ) interval, involving the atrial depolarization and the conduction via the atrioventricular junction and the His-Purkinje system	short: ≤121 ms for women and ≤129 ms for men; long: >200 ms
PR (PQ) interval variation	The difference between the maximal and minimal PR (PQ) interval on the standard 12-lead ECG	>36.5 ms
**ECG Parameters of Ventricular Electrical Activity**	**Description**	**Cut-Off Values Predicting AF**
LVH	left ventricular hypertrophy	presence
PVC, NSVT	premature ventricular contractions, non-sustained ventricular tachycardia	presence
ST-T abnormalities		presence
QTc	corrected QT interval	prolongation
BBB	bundle branch block	presence

ms = millisecond, μV = microvolt, mVs = millivolt multiplied by second.

**Table 2 medicina-56-00410-t002:** Interatrial conduction block (IAB) types, their electrophysiological background and the corresponding ECG features.

IAB Type	Electrophysiological Background	ECG Features
first-degree (partial)	delayed conduction in the zone of BB	P-wave duration >120 ms
third-degree (advanced)	blocked conduction in the zone of BB: caudocranial, retrograde activation of the left atrium from the low right atrium (coronary sinus and to a lesser degree, the fossa ovalis)	P wave duration >120 ms with biphasic morphology (a positive initial component and a terminal negative component) in the inferior leads (II, III, aVF)
second-degree	delayed or blocked conduction in the zone of BB	transient appearance of first-degree and/or third-degree IAB pattern on the same ECG recording (atrial aberrancy)—related or not to an atrial premature beat

**Table 3 medicina-56-00410-t003:** The predictive value of IAB for AF occurrence in different clinical settings and in population studies.

Predicting Value	Clinical Setting/Study	Reference	Observations
IAB as a predictor of new onset AF	- patients with Chagas cardiomyopathy and ICDs	[21]	
	- patients with NSTEMI	[22]	
	- post-transcatheter aortic valve replacement	[23]	
	- patients with severe heart failure undergoing cardiac resynchronization device implantation and no AF history	[24]	IAB proved to be an independent predictor of AF
IAB as a predictor of AF recurrence	- post-cardioversion	[25]	
	- post-pulmonary vein isolation	[26]	
	- post-atrial flutter ablation (cavotricuspid isthmus ablation)	[27]	
IAB as a predictor of AF occurrence in the general population	- the Atherosclerosis Risk in Communities (ARIC) study	[28]	risk factors associated with advanced IAB development identified by the ARIC study were age, white race, male gender, body mass index, systolic blood pressure, use of antihypertensive medication, low-density lipoprotein cholesterol
	- the Copenhagen ECG Study	[29]	

ICD = implantable cardioverter defibrillator; NSTEMI = non ST-segment elevation myocardial infarction.

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
