# Peer review of "Bayés’ Syndrome—A Comprehensive Short Review"

_medicina, 2020, doi:10.3390/medicina56080410_

Round 1

Reviewer 1 Report

The manuscript is interesting and well written.

I have only minor comments.

  1. Please add precordial leads to Figures 1, 2, and 3. The ECG in figure 3 looks like left anterior fascicular block.
  2. I think that inter-atrial block may be associated with sinus node dysfunction and atrio-ventricular block because atrial remodeling may advance in the whole atrium. In this regard, pacemaker implantation may be required more commonly. Please comment.
  3. Are there any differences in the prevalence and characteristics of inter-atrial block between genders and between races?
  4. LAE is not a common abbreviation.
  5. Are there any reports demonstrating iatrogenic inter-atrial block after catheter ablation or valvular surgeries?
  6. Please provide more detailed comments regarding the inter-atrial tracts (small myocardial fibers) other than Bachmann’s bundle.

Reviewer 2 Report

Thank you for giving me the opportunity to review the paper. The author mentioned the comprehensive review of Bayés syndrome. The whole part is well-written. The manuscript is useful for understanding the perspective.  It is very helpful for clinical practice. I think that the manuscript is eligible for publication.

Author Response

Thank you very much for your opinion about our manuscript.

The English language and style has been improved.